# How Far Are We from the Planetary Health Diet? A Threshold Regression Analysis of Global Diets

**DOI:** 10.3390/foods11070986

**Published:** 2022-03-28

**Authors:** Yifan Chen, Li Chai

**Affiliations:** 1International College Beijing, China Agricultural University, Beijing 100083, China; 2019314050211@cau.edu.cn; 2Agricultural College, Oklahoma State University, Stillwater, OK 74078-1015, USA; 3College of Economics and Management, China Agricultural University, Beijing 100083, China

**Keywords:** threshold regression, diet, food consumption, Planetary Health Diet, food system

## Abstract

Global diets and food system not only influence human health conditions but also have a great effect on environmental sustainability. The Planetary Health Diet (PHD) proposed by the Lancet Commission is considered as a sustainable diet that meets human’s nutritional demands yet poses less pressure on the environment. In this study, we examine how the economic condition, i.e., Gross Domestic Product per capita (GDP per capita), affects the deviations of current diets from the PHD at the country level by using a threshold regression model. The results show three dimensions regarding food consumption patterns in all 11 kinds of foods across the globe, as evidenced from the data in 147 countries as of 2018. First, the findings indicate that there exist deviations from the PHD for all kinds of foods, which could guide policymakers to make dietary improvements. Second, we find that GDP per capita impacts food consumption patterns with all kinds of foods. The results demonstrate that the changing rates of food consumption amounts decrease as the GDP per capita increases. Finally, we calculate the GDP per capita thresholds for all kinds of foods, and we find the number of thresholds ranging from zero to two. Specifically, 20,000 PPP (current international $), the GDP per capita boundary distinguishing developing and developed countries, is the first GDP per capita threshold influencing the food consumption amount. What is more, the second GDP threshold is 40,000 PPP (current international $), which is the average GDP per capita of developed countries. Thus, we identify the countries that require more financial assistance from a GDP per capita perspective.

## 1. Introduction

Global diets and food systems [1], and the populations relying on them, are experiencing both health and sustainability challenges. When it comes to the world’s disease burden, low-quality diet is the largest cause [2]. There are approximately 800 million people worldwide undernourished, while there are around 2 billion adults overweight or obese and more than 2.5 billion people worldwide suffer from malnutrition [3]. The current global food system is estimated to contribute to 11 million preventable adult deaths per year. Worse still, climatic balance and ecosystem adaptability are also threatened by the current global food system [4]. It is known that as climate change affects food systems, food systems also affect climate change. It is estimated that about 1/4 of all greenhouse gas emissions may be caused by the global food system [5]. Moreover, the global food system affects 70% of freshwater use [6], 40% of land use [7], and 78% of eutrophication [8]. Together, this makes agriculture and food production the number one factor of environmental damage [9], which has a great effect on human and planetary health [4,10].

Thus, a great food system transformation is an urgent need. The ultimate goal of the transformation is to form a sustainable diet, which is essentially considered to be a diet that has low environmental impact while contributing to food security and meeting the health and nutritional needs of current and future generations [4]. This is to say, in order to provide sufficient food to meet nutritional needs for the projected 2050 global population of 10 billion people and stay within the limits of the planet’s resources at the same time [2], how food systems operate must change, including which food is consumed and by whom [2]. The Food and Agriculture Organization of the United Nations (FAO) [11] agreed with the need to transition to a more healthy and sustainable diet.

This study uses a threshold regression model to examine how GDP perc capita affects the deviations of current diets from the Planetary Health Diet (PHD), which is a kind of sustainable diet proposed by Willett et al. in the Lancet Commission report [4]. The Lancet Commission’s report [4] is the first to comprehensively integrate the needs of human health with planetary sustainability principles into a single set of global dietary recommendations. The planetary boundaries framework proposes a safe operating space to design the PHD. The cropland use, biodiversity loss, water use, greenhouse-gas emissions, and nitrogen and phosphorus pollution that can be due to food production are considered. Meanwhile, the PHD allows great flexibility because they are compatible with a wide variety of foods, agricultural systems, cultural traditions, and individual dietary preferences. These elements can be combined in various types of omnivore, vegetarian, and vegan diets.

The PHD mainly focuses on 11 kinds of foods: grains, tubers or starchy vegetables, vegetables, fruits, dairy foods, red meat, poultry, eggs, seafood, legumes, and nuts. It sets the optimal intake of food groups for these 11 kinds of foods (i.e., 200 g/day/capita of fruit). The PHD is rich in micronutrient-rich foods such as fruits, vegetables, whole grains, legumes, nuts, and unsaturated oils. It includes moderate levels of seafood and poultry and very little red meat, refined grains, and added sugars. Such a diet amounts to an intake of 2500 kilocalories per person per day [4]. It shows that the intake of most nutrients increased after the adoption of this diet compared with current consumption patterns, except for vitamin B12, which needs fortification or supplementation [4,10,12,13,14].

There has not been any research to examine the deviations of current diets from sustainable diets. How the PHD diverges from existing dietary guidance is an open question, since most food-based dietary guidelines have been developed without reference to environmental sustainability [15]. There is also a lack of research on how GDP per capita affects food consumption on the country level, although some researchers have recognized that economic development is one of the factors that determine food choice [16]. It is found that a shift to the PHD requires that the necessary foods be both available and affordable for countries with low GDP per capita [17]. Otherwise, people in poor countries cannot adopt the PHD. Noticeably, it has been shown that there are significant variations in the affordability of healthy diets between low, middle, and high-income countries [18]. The PHD is affordable for high-income countries but unaffordable for low-income countries [19]. Rosergrant et al. analyze the implications for future global cereal and meat supply and demand resulting from changes in global income [20]. Gerbens-Leenes et al. [21] explore general relationships between economic change and the rate of change in food consumption patterns. They find nutrition transition began in developed countries 300 years ago, which coincided with great economic growth [22]. The food consumption pattern has very important implications for world food demand and world agriculture [23]. Thus, there is a need to examine how the economic condition, i.e., per capita Gross Domestic Product (GDP per capita), affects the deviations of current diets from the Planetary Health Diet. What is more, GDP per capita thresholds should be observed to guide policymakers.

The main hypothesis of this study is the threshold effects of GDP per capita on food consumption. This study introduces the Planetary Health Diet and finds the deviations of current diets from the PHD at the country level. This study also examines how GDP per capita affects food consumption using a threshold regression model. Moreover, this study calculates the GDP per capita thresholds for all kinds of foods. Based on the findings, we guide to make dietary improvements and identify the countries that require more financial assistance from a GDP per capita perspective.

The remainder of the paper is organized as follows. In the next section, we discuss our data sources and the variables. In Section 3, we describe our threshold regression model. Section 4 reports the results of the threshold regression model. Section 5 discusses the findings. Finally, Section 6 concludes the paper.

## 2. Data

### 2.1. Data Source

We collect data from 147 countries in 2018 and build a threshold regression model. To illustrate how GDP per capita influences food consumption amount more clearly, we highlight seven countries: the Central African Republic, Ethiopia, India, Brazil, China, the United Kingdom, and the Unites States. For the rest of the world, we also mark them. The reason why the seven countries are chosen are as follows. Two developed countries are chosen from two different continents: the United Kingdom and the Unites States; two developing countries are chosen from two different continents: Brazil and China; two countries that have the lowest GDP per capita are chosen: the Central African Republic and Ethiopia; two countries that have the most population are chosen: India and China.

The consumption quantities (kg/capita/year) data of every kind of food are obtained from the 2018 Food Balance Sheet in the FAO database. The consumption quantities (g/capita/day) of each kind of food are obtained by timing the consumption quantities (kg/capita/year) data by 2.74 (1000 g/365 days). The classification of 11 food groups is shown in Table 1. The GDP per capita, PPP (current international $) is obtained from the World Bank database. The total agricultural area is obtained from the FAO database, and the total population is obtained from the UN database. The agricultural land is the sum of cropland and land used as pasture for grazing livestock. The population aged over 65 and the total population is obtained from the UN database. The Human Development Index (HDI) is obtained from the UN database.

### 2.2. Variables

Total score (>1), total score (<1), and the consumption quantities (g/capita/day) of each of 11 kinds of food are the explained variables. The total score (>1) and the total score (<1) show the overall picture of the deviation of current diets from the PHD at the country level, and the methodology for calculating them is shown below. Firstly, we calculate the score of every kind of food—the quotient of the current consumption amount and the recommendation of the PHD. Secondly, we subtract one from the scores of every kind of food to obtain the differences. Then, to calculate the scores of the foods that are more than the recommendation of the PHD (the differences are positive), we take the sum of the positive differences. This is named total score (>1). Total score (>1) measures the degree of adequate consumption or even excessive food consumption. For each country, to calculate the scores of the foods that are less than the recommendation of the PHD (the differences are negative), we take the sum of the negative differences. This is named total score (<1). Total score (<1) measures the degree of inadequate food consumption.

The threshold variable and explanatory variable in this model is GDP per capita, PPP (current international $), which shows a country’s GDP divided by its total population. GDP at Purchasing Power Parity (PPP) per capita is used because PPP takes not only the exchange rate but also the relative cost of living into account. Therefore, PPP provides a more accurate picture of the real differences in income.

The control variables in this model are the agricultural land per capita, the level of population aging, and the human development index (HDI). The agricultural land area per capita shows a country’s total agricultural land divided by its total population. The proportion of people aged over 65 is used to represent the level of population aging. The HDI is a summary measure of average achievement in key dimensions of human development: a long and healthy life, being knowledgeable, and having a decent standard of living.

## 3. Empirical Strategy

### 3.1. Basic Structure of the Threshold Regression Model

Based on the panel data threshold model of Wang [24], this paper sets the threshold regression model. Accounting for the fact that we focus on 2018 data, we build a cross-section data threshold model. Instead of only considering a single threshold, multiple thresholds appear in the analysis. The model form of the threshold value is as follows:(1)$$\begin{array}{c} FC_{i}=\alpha_{i}+\beta_{1}*GDP_{i}*I\left(GDP_{i}\leq r_{1}\right)+\beta_{2}*GDP_{i}*I\;\left(r_{1}<GDP\leq r_{2}\right)\\ +\ldots +\beta_{n}*GDP_{i}*I\left(GDP_{i}>r_{n}\right)+ZP+e_{i} \end{array}$$
where subscript i represents the country. $FC_{i}$ represents the explained variable, $\alpha_{i}$ represents the constant term, $\beta_{i}$ represents the coefficient, $GDP_{i}$ is the explanatory variable and the threshold variable, r represents the threshold value, $I\;\left(\cdot \right)$ represents an illustrative function, Z is a 1 × 3 matrix $\left[\begin{array}{ccc} A_{i} & B_{i} & C_{i} \end{array}\right]$, representing the control variables, P is a 3 × 1 matrix $\left[\begin{array}{c} \;\rho_{1}\;\\ \rho_{2}\\ \rho_{3} \end{array}\right]$, representing the coefficients of the control variables, and $e_{i}$ represents the random error.

### 3.2. Existence Test

Firstly, a significance test is used to test the existence of the first threshold. We use the LM (Lagrange multiplier) test. The null hypothesis is:(2)$$H_{0}:\;m_{1i}=m_{2i}\;\left(i=0,1,2\right)$$

If $H_{0}$ hypothesis is rejected, it indicates that there are significant changes between two sections (i.e., there is a threshold). Otherwise, there is no threshold.

The tests for the second and more thresholds are similar to the test for the first threshold. Thus, the number of thresholds is confirmed.

### 3.3. Estimating Estimation of Threshold

To estimate r in Equation (1), one can search over a subset of the threshold variable, $GDP_{i}.$ r’s estimator is the value that minimizes the residual sum of squares RSS.

### 3.4. Authenticity Test

One way to test $r=r_{0}$ is to form the confidence interval using the likelihood-ratio (LR) statistic, as follows:(3)$${\mathrm{LR}}_{1}\left(r\right)=\{{\mathrm{LR}}_{1}\left(r\right)-\frac{{\mathrm{LR}}_{1}(\hat{r)}\}}{\left({\hat{\sigma }}^{2}\right)\;}$$
(4)$$\mathrm{Pr}\left(x<\xi \;\right)={\left(1-e^{-\frac{x}{2}}\right)}^{2}$$
where $\hat{r}$ is a consistent estimator for r.

Given significance level α, the α quantile can be computed from the following formula:(5)$$c\left(\alpha \right)=-2\log\left(1-{\left(1-\alpha \right)}^{\frac{1}{2}}\right)$$

If LR1($r_{0}$) exceeds $c\left(\alpha \right)$, then we reject $H_{0}$.

## 4. Results

The results of the threshold regression model are summarized in Table 2, which is followed by a detailed analysis of the results.

As shown in Figure 1, total score (<1) has one GDP per capita threshold: $31,467. The slope of the first region is $1.0*10^{-4}$ and the slope of the second region is $4.2*10^{-6}$. However, the tendency is close to 0. The rate nearing 0 decreases as GDP per capita increases. Total score (>1) has one GDP per capita threshold: $29,456. The slope of the first region is $1.9*10^{-4}$, and the slope of the second region is $1.5*10^{-6}$. When GDP per capita is less than $29,456, the total score (>1) increases quickly as GDP per capita increases. When GDP per capita is greater than $29,456, the total score (>1) increases much more slowly and is almost flat.

As for the Central African Republic, Ethiopia, India, Brazil and China, their total scores (both >1 and <1) increase quickly as GDP per capita increases. As for the United Kingdom and United States, their total scores (both >1 and <1) increase much more slowly as GDP per capita increases.

In conclusion, when the GDP per capita is smaller, the score changes quickly. When GDP per capita is greater, the score changes slowly. Ethiopia, India, Brazil, China, the United Kingdom, and the United States fit this rule. However, the Central African Republic’s total score (>1) is too large. The reason is that people in the Central African Republic consume too much tuber or starchy vegetables.

The 11 kinds of foods can be divided into three groups: foods that are encouraged to consume (vegetables, fruits, dairy food, nuts), foods that should be consumed in appropriate amounts (tubers or starchy vegetables, grains, legumes, poultry, eggs and seafood), and foods that are discouraged to consume (red meat).

Almost each of the 11 kinds of foods has GDP per capita thresholds. China and Brazil, whose GDP per capita is around $20,000, are usually close to the threshold. In addition, the threshold is also a boundary distinguishing developing countries and developed countries. What is more, 40,000 PPP (current international $), the average GDP per capita of developed countries, is also a GDP per capita threshold influencing the consumption amounts of foods.

### 4.1. Vegetables

According to the PHD, the recommended consumption amount is 600 g/capita/day for vegetables. As shown in Figure 2a, vegetables have one GDP per capita threshold: $18,064. The slope of the first region is $2.0*10^{-2}$, and the slope of the second region is $4.2*10^{-4}$. When GDP per capita is less than $18,064, the consumption amount (g/capita/day) of vegetables increases quickly as GDP per capita increases. When GDP per capita is greater than $18,064, the consumption amount (g/capita/day) of vegetables is almost constant as GDP per capita increases, fluctuating around 300. As for Central African Republic, Ethiopia, India, Brazil and China, their vegetables consumption quantities (g/capita/day) increase quickly as GDP per capita increases. As for the United Kingdom and United States, their vegetables consumption quantities (g/capita/day) increase much more slowly as GDP per capita increases. China has a very high vegetable consumption amount (g/capita/day).

### 4.2. Fruits

According to the PHD, the recommended consumption amount is 200 g/capita/day for fruits. As shown in Figure 2b, fruits have one GDP per capita threshold: $18,296. The slope of the first region is $9.1*10^{-3}$, and the slope of the second region is $-1.6*10^{-4}$. When the GDP per capita is less than $18,296, the consumption amount (g/capita/day) of fruits increases quickly as GDP per capita increases. When GDP per capita is greater than $18,296, the consumption amount (g/capita/day) of fruits is almost constant as GDP per capita increases, fluctuating around 200. As for the Central African Republic, Ethiopia, India, Brazil, and China, their fruits consumption quantities (g/capita/day) grow quickly as the GDP per capita increases. As for the United Kingdom and United States, their fruits consumption quantities (g/capita/day) increase much more slowly as GDP per capita increases.

### 4.3. Dairy Food

According to the PHD, the recommended consumption amount is 250 g/capita/day for dairy food. As shown in Figure 2c, dairy food does not have a GDP per capita threshold. The slope of increasing is $7.8*10^{-3}$. The dairy food consumption quantities (g/capita/day) of Central African Republic, Ethiopia, India, Brazil, China, the United Kingdom and the United States increase as the GDP per capita increases. 

### 4.4. Nuts

According to the PHD, the recommended consumption amount is 25 g/capita/day for nuts. As shown in Figure 2d, nuts do not have GDP per capita threshold. The slope of increasing is $1.2*10^{-7}$, which is almost flat and fluctuates around 12.5. The nuts consumption quantities (g/capita/day) of Central African Republic, Ethiopia, India, Brazil, China, the United Kingdom, and the United States are almost the same.

### 4.5. Tubers or Starchy Vegetables

According to the PHD, the recommended consumption amount is 50 g/capita/day for tubers or starchy vegetables. As shown in Figure 3a, tubers or starchy vegetables have two GDP per capita thresholds: the first one is $12,206 and the second one is $48,136. The slope of the first region is $-2.7*10^{-2}$, the slope of the second region is $-2.1*10^{-3}$, and the slope of the third region is $-3.5*10^{-4}$. The consumption amount (g/capita/day) of tubers or starchy vegetables decreases dramatically; then, the decreasing rate slows down and finally flattens. When the GDP per capita is less than $12,206, the consumption amount (g/capita/day) of tubers or starchy vegetables decreases quickly as the GDP per capita increases. When the GDP per capita is between $12,206 and $48,136, the decreasing rate of the consumption amount (g/capita/day) of tubers or starchy vegetables slows down as the GDP per capita increases. When the GDP per capita is greater than $48,136, the consumption amount (g/capita/day) of tubers or starchy vegetables is almost constant as the GDP per capita increases, fluctuating around 125. The Central African Republic, Ethiopia, and India are in the first region. Brazil and China are in the second region. The United Kingdom and United States are in the third region. The grouping situation is almost the same for grains and tuber or starchy vegetables.

### 4.6. Grains

According to the PHD, the recommended consumption amount is 232 g/capita/day for grains. As shown in Figure 3b, grains have two GDP per capita thresholds: the first one is $15,992 and the second one is $48,756. The slope of the first region is $4.9*10^{-3}$, the slope of the second region is $6.2*10^{-4}$, and the slope of the third region is $-6.7*10^{-4}$. The growth rate slows down, and finally, the consumption amount (g/capita/day) of grains starts to decrease a little. When the GDP per capita is less than $15,992, the consumption amount (g/capita/day) of grains increases as the GDP per capita increases. When the GDP per capita is between $15,992 and $48,756, the growth rate of the consumption amount (g/capita/day) of grains slows down as GDP per capita increases. When the GDP per capita is greater than $48,756, the consumption amount (g/capita/day) of grains decreases as the GDP per capita increases. The Central African Republic, Ethiopia, and India are in the first region. Brazil and China are in the second region. The United Kingdom and United States are in the third region. The grains consumption amount (g/capita/day) of the Central African Republic is very low.

### 4.7. Legumes

According to the PHD, the recommended consumption amount is 100 g/capita/day for legumes. As shown in Figure 3c, legumes have one GDP per capita threshold: $12,206. The slope of the first region is −$1.9*10^{-3}$, and the slope of the second region is $-1.6*10^{-5}$. The decreasing rate slows down and finally flattens. When the GDP per capita is less than $12,206, the consumption amount (g/capita/day) of legumes decreases quickly as the GDP per capita increases. When the GDP per capita is greater than $12,206, the consumption amount (g/capita/day) of legumes is almost constant as the GDP per capita grows, fluctuating around 10. Compared with the Central African Republic, Ethiopia, India, Brazil, and China, the legumes consumption quantities (g/capita/day) of the United Kingdom and United States decrease much slowly as the GDP per capita increases.

### 4.8. Poultry

According to the PHD, the recommended consumption amount is 29 g/capita/day for poultry. As shown in Figure 3d, poultry has one GDP per capita threshold: $26,167. The slope of the first region is $3.6*10^{-3}$, and the slope of the second region is $-2.6*10^{-5}$. The growth rate slows down and finally flattens. When the GDP per capita is less than $26,167, the consumption amount (g/capita/day) of poultry increases quickly as the GDP per capita increases. When the GDP per capita is greater than $26,167, the consumption amount (g/capita/day) of poultry is almost constant as the GDP per capita increases, fluctuating around 72.5. As for the Central African Republic, Ethiopia, India, Brazil, and China, their poultry consumption quantities (g/capita/day) increase quickly as the GDP per capita increases. As for the United Kingdom and United States, their poultry consumption quantities (g/capita/day) increase much more slowly as the GDP per capita increases. The consumption amount (g/capita/day) of poultry in the United States is relatively high.

### 4.9. Eggs

According to the PHD, the recommended consumption amount is 13 g/capita/day for eggs. As shown in Figure 3e, eggs have one GDP per capita threshold: $8866. The slope of the first region is $2.1*10^{-3}$, and the slope of the second region is $2.4*10^{-4}$. The growth rate slows down and finally grows at a small rate. When the GDP per capita is less than $8866, the consumption amount (g/capita/day) of eggs increases quickly as the GDP per capita increases. When the GDP per capita is greater than $8866, the consumption amount (g/capita/day) of the eggs growth rate slows down as the GDP per capita increases. As for the Central African Republic, Ethiopia, and India, their eggs consumption quantities (g/capita/day) increase quickly as the GDP per capita increases. As for Brazil, China, the United Kingdom, and the United States, their eggs consumption quantities (g/capita/day) increase much more slowly as the GDP per capita increases. The consumption amount (g/capita/day) of eggs in China is relatively high. 

### 4.10. Seafood

According to the PHD, the recommended consumption amount is 28 g/capita/day for seafood. As shown in Figure 3f, seafood does not have a GDP per capita threshold. The increasing slope is $8.0*10^{-4}$. The seafood consumption quantities (g/capita/day) of the Central African Republic, Ethiopia, India, Brazil, China, the United Kingdom, and the United States increase as the GDP per capita increases. China has a relatively high seafood consumption amount (g/capita/day).

### 4.11. Red Meat

According to the PHD, the recommended consumption amount is 14 g/capita/day for red meat. Red meat does not have a GDP per capita threshold. The increasing slope is $1.96*10^{-3}$. The meat consumption quantities (g/capita/day) of the Central African Republic, Ethiopia, India, Brazil, China, the United Kingdom, and the United States increase as the GDP per capita increases. The relationship between GDP per capita and red meat consumption is shown in Figure 4.

## 5. Discussion

When it comes to the foods that the government should encourage people worldwide to consume, vegetables, fruits, dairy food, and nuts stand out. Governments should encourage the consumption of vegetables and fruits. Vegetables and fruits are important vitamin and mineral sources. The trend of dairy food is gratifying: the consumption amount of dairy food increases as the GDP per capita increases. Dairy food is an important source of protein and minerals. However, people worldwide do not consume enough nuts. The consumption amount of nuts remains almost unchanged as the GDP per capita increases. The government should encourage the consumption of nuts. Nut consumption has been associated with several health benefits, such as antioxidant, hypocholesterolemic, anticancer, and anti-inflammatory benefits. In addition, fruits and dairy food can reduce the risk of T2D and colorectal cancer [25,26]. Fruits, vegetables, and nuts can reduce the risk of cardiovascular diseases, such as coronary heart disease, stroke, and heart failure [26,27].

Grains, tubers or starchy vegetables, legumes, dairy foods, poultry, eggs, and seafood are foods that people should consume in moderation. Both grains and tubers or starchy vegetables provide carbohydrates for humans. As people become richer (GDP per capita increases), they give up tubers or starchy vegetables and turn to grains for carbohydrates. Governments should prevent people from eating too many tubers or starchy vegetables, which are inferior and innutritious starch sources. Eating boiled potatoes, such as French fries, can lead to T2D and hypertension [28]. However, people should also consume grains in moderation. Although whole grains can reduce the risk of chronic disease and premature mortality, a positive association between consumption of refined grains and T2D may exist [29,30].

Governments should encourage people to consume legumes, poultry, eggs, and seafood in moderation, which are sources of proteins. Legumes are good substitute sources of proteins if people cannot afford dairy food and eggs. However, legumes are common allergens and are high in uric acids, which makes people with gout unable to eat them. Compared with the proteins in meat, which are nutritionally complete and contain all the essential amino acids people need for good health, legumes usually lack one or more of the essential amino acids. Thus, relying merely on legumes to meet protein requirements might result in a nutritional deficiency. As people become richer (GDP per capita increases), they turn to better protein sources. However, T2D risk increases with the increasing consumption of eggs [31]. Similarly, poultry is significantly associated with incident CVD [32,33,34]. As people become richer (GDP per capita increases), the consumption amount of seafood also increases stably. Nevertheless, the government should also prevent people from eating too much seafood. Contamination of freshwater fish with toxic heavy metals, such as mercury, and metalloids makes exceedances of benchmarks likely to have deleterious health effects on people [35].

People are discouraged from eating red meat. It is worrying to see the trend of red meat: the consumption amount of red meat increases as the GDP per capita increases. Red meat can increase the risk of heart diseases, cause hardening of blood vessels, lead to cancer, and cause T2D and inflammation. Processed red meat is even worse [35,36].

Lancet’s projections [4] indicate that food production could increase greenhouse gas emissions, cropland use, freshwater use, and nitrogen and phosphorus application by 50–90% from 2010 to 2050 in absence of dedicated mitigation measures. This increase would push key biophysical processes beyond the boundaries and safe operating space for food production. Different food groups affect the environment to different extents: animal source foods are responsible for about three-quarters of climate change effects, whereas staple crops, such as wheat, rice, and other cereals, are responsible for one-third to one-half of pressures on other environmental domains. Seafood is a particularly diverse food category, and environmental effects can differ substantially.

The reason why we introduce the PHD is its potential to meet human nutritional demands and pose less pressure on the environment. Compared with Fischer and Garnett [15] and Rosegrant et al. [20], we show the current food system in every country and the deviation of current diets from the PHD at the country level. What is more, we divide the 11 kinds of foods into three groups: foods that should be encouraged to consume (vegetables, fruits, dairy food, nuts); foods that should be consumed in appropriate amounts (tubers or starchy vegetables, grains, legumes, poultry, eggs, and seafood); foods that should be discouraged to consume (red meat). Thus, governments are better equipped with tools to improve the local food system. Countries should be alarmed if they lack foods whose consumption is encouraged or they exceed the recommended consumption amounts of foods whose consumption is discouraged. For example, almost all countries consume too few nuts and too much meat.

It is also indicated in our results that different kinds of food respond differently to increasing the GDP per capita. Among the 11 kinds of food, the consumption amounts of tubers or starchy vegetables and legumes decrease as the GDP per capita increases, while the consumption amounts of other kinds of foods increase as the GDP per capita increases. Tubers or starchy vegetables are inferior sources of calories, and legumes are inferior sources of proteins. We prove that the diets of the poorest people tend to be composed principally of cheap starchy staple foods: wheat, rice, potatoes, cassava, yams, millet, sweet potatoes, and the like [37,38]. However, the direct per capita food consumption of maize and coarse grains is declining, as with increasing GDP per capita, consumers shift to wheat and rice. Based on the findings of Rask [39] and Yotopoulos [40], a central feature of food demand under GDP per capita growth is the shift from reliance on direct consumption of grains and other starchy staples into more diversified diets including edible oils and protein-rich animal products. Similar to Bennett’s law, we find an increase in spending on fruits and vegetables and animal-based products as the GDP per capita increases [41]. We also point out that the growth in per capita meat and cereal consumption in developed countries has slowed dramatically, as these countries have already reached very high levels of meat consumption in the past decades [20].

Furthermore, the results also prove that that are general relationships between economic condition and the food consumption pattern, which is supported by Gerbens-Lennes et al. in 2010 [21]. Based on the findings of Hirvonen et al. [19], we further explore the threshold effects of GDP per capita on food consumption and calculate the GDP per capita thresholds for all kinds of foods. Our results of the threshold regression model indicate that $20,000, which is the boundary distinguishing developing and developed countries, is almost a threshold for each of 11 kinds of foods. Thus, international organizations should raise funds to carry out an aid program to help developing countries, especially those whose GDP per capita is less than $20,000. Through these aid programs, people in developing countries can have enough foods to consume and even turn to better nutrient resources. What is more, the aids to countries whose GDP per capita is less than $20,000 can achieve notable results, for the consumption quantities change relatively quickly in these countries as the GDP per capita changes. For developed countries, international organizations should encourage them to increase the replacement of animal source foods with plant-based foods for the benefit of human health and sustainability.

## 6. Conclusions

Global diets and food system not only influence human health conditions but also have a great effect on environmental sustainability. This study introduces the Planetary Health Diet (PHD) proposed by the Lancet Commission, which is a kind of sustainable diet that meets human’s nutritional demand but poses less pressure on the environment. This study examines how GDP per capita affects the deviations of current diets from PHD at the country level by using a threshold regression model based on data collected from 147 countries in 2018. 

Compared with Fischer and Garnett [15] and Rosegrant et al. [20], we show the current food system in every country and the deviation of current diets from the PHD at the country level. Furthermore, we provide evidence that there are general relationships between economic conditions and the food consumption patterns, which is supported by Gerbens-Lennes et al. in 2010 [22]. Based on Hirvonen et al. [17], we further explore how the GDP per capita affects the deviations of current diets from the PHD using the threshold regression model and calculate the GDP per capita thresholds for all kinds of foods. Our study identifies the countries that require more financial assistance from a GDP per capita perspective. The results are informative for policymakers to implement appropriate interventions to make dietary improvements.

This study mainly explores whether people consume enough of every kind of food and the threshold effects of GDP per capita on food consumption amounts. Instead of focusing on each individual country, this study focuses on how GDP per capita influences the trend of food consumption across the globe. Thus, the influences of tradition and culture on food consumption deserve further exploration, which is something we leave for future research. Researchers are also encouraged to establish an evaluation system to measure the quality of the current food system in every country. Moreover, researchers may explore the feasibility of the PHD by taking the food costs and inequity into consideration.

## Figures and Tables

**Figure 1 foods-11-00986-f001:**
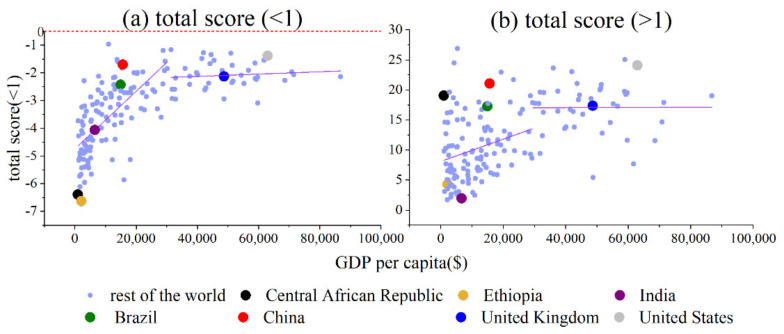
GDP per capita vs. (**a**) total score (<1) and (**b**) total score (>1).

**Figure 2 foods-11-00986-f002:**
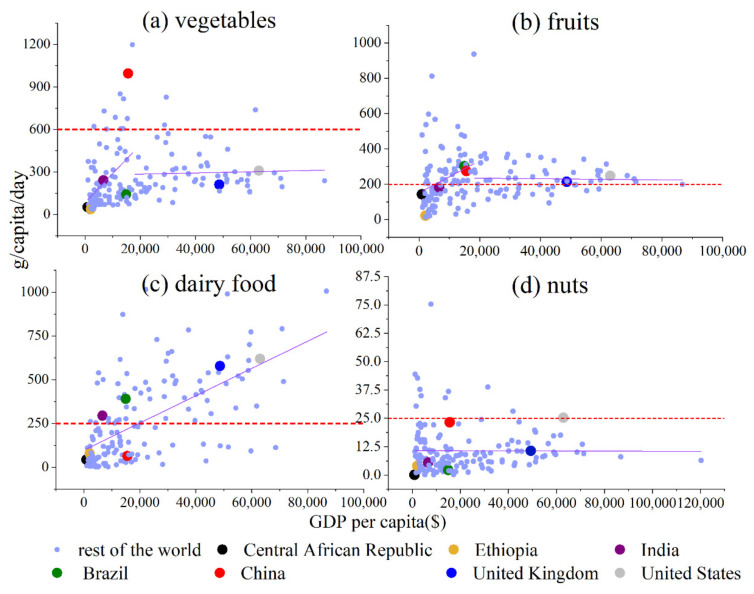
GDP per capita vs. food consumption amounts of (**a**) vegetables, (**b**) fruits, (**c**) dairy foods, and (**d**) nuts (those foods are encouraged to consume).

**Figure 3 foods-11-00986-f003:**
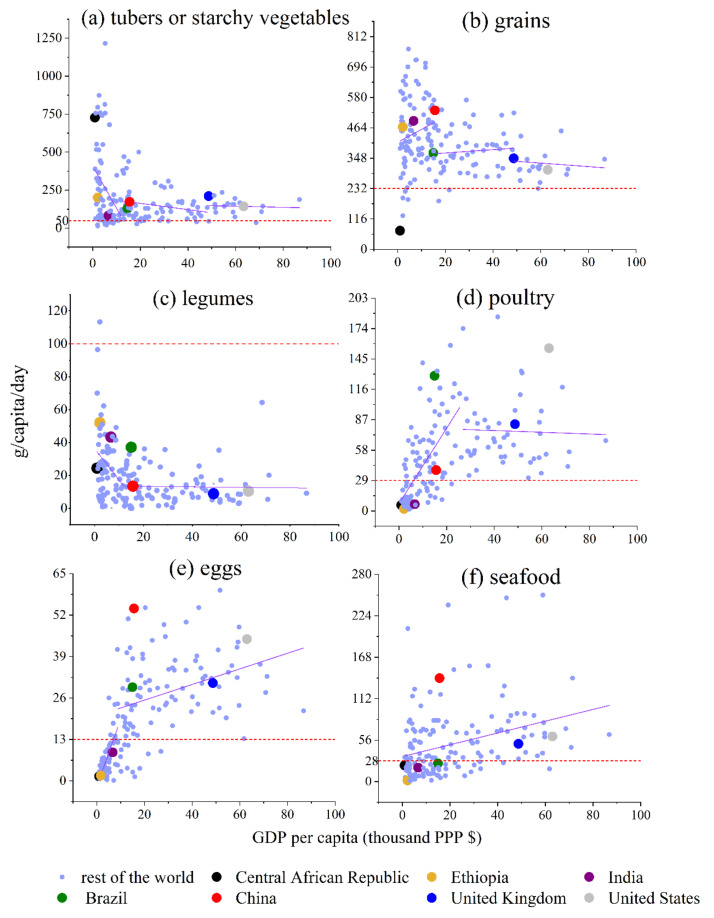
GDP per capita vs. food consumption amounts of (**a**) tubers or starchy vegetables, (**b**) grains, (**c**) legumes, (**d**) poultry, (**e**) eggs, and (**f**) seafood (those foods should be consumed in moderation).

**Figure 4 foods-11-00986-f004:**
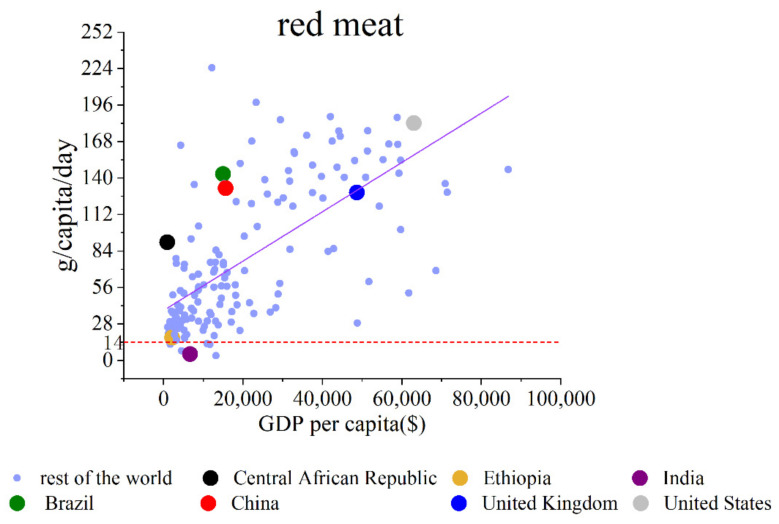
GDP per capita vs. food consumption amount of red meat (this food is discouraged to consume).

**Table 1 foods-11-00986-t001:** The classification of 11 food groups.

Food Groups	Specific Kind of Group	PHD Recommendations (g/Capita/Day)
grains	Wheat and products; Rice and products; Barley and products; Maize and products; Rye and products; Oats; Millet and products; Sorghum and products; Cereals, Other	232
tubers	Cassava and products; Potatoes and products; Sweet potatoes; Yams; Roots, Other	50
vegetables	Onions; Tomatoes and products; Vegetables, Other	600
fruits	Apples and products; Bananas; Citrus, Other; Coconuts—Incl Copra; Dates; Fruits, Other; Grapefruit and products; Grapes and products (excl wine); Lemons, Limes and products; Oranges, Mandarins; Pineapples and products; Plantains	200
dairy foods	Butter, Ghee; Cream; Milk—Excluding Butter	250
red meat	Bovine Meat; Meat, Other; Mutton and Goat Meat; Pig Meat	14
poultry	Poultry Meat	29
eggs	Eggs	13
seafood	Aquatic Animals, Others; Aquatic Plants; Cephalopods; Crustaceans; Demersal Fish; Fish, Body Oil; Fish, Liver Oil; Freshwater Fish; Marine Fish, Other; Meat, Aquatic Mammals; Mollusks, Other; Pelagic Fish	28
legumes	Beans; Peas; Pulses, Other and products	100
nuts	Cottonseed; Groundnuts; Nuts and products; Rape and Mustard seed; Soybeans; Sunflower seed	25

**Table 2 foods-11-00986-t002:** The slopes and the thresholds of the threshold regression model.

	Food Groups	Slope 1	Threshold 1	Slope 2	Threshold 2	Slope 3
total scores	total score (<1)	$1.0*10^{-4}$	$31,467	$4.2*10^{-6}$		
	total score (>1)	$1.9*10^{-4}$	$29,456	$1.5*10^{-6}$		
encouraged	vegetables	$2.0*10^{-2}$	$18,064	$4.2*10^{-4}$		
	fruits	$9.1*10^{-3}$	$18,296	$-1.6*10^{-4}$		
	dairy foods	$7.8*10^{-3}$				
	nuts	$1.2*10^{-7}$				
moderate	tubers	$-2.7*10^{-2}$	$12,206	$-2.1*10^{-3}$	$48,136	$-3.5*10^{-4}$
	grains	$4.9*10^{-3}$	$15,992	$6.2*10^{-4}$	$48,756	$-6.7*10^{-4}$
	legumes	$-1.9*10^{-3}$	$12,206	$-1.6*10^{-5}$		
	poultry	$3.6*10^{-3}$	$26,167	$-2.6*10^{-5}$		
	eggs	$2.1*10^{-3}$	$8866	$2.4*10^{-4}$		
	seafood	$8.0*10^{-4}$				
discouraged	red meat	$1.96*10^{-3}$				

## Data Availability

Not applicable.

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
