# Peer review of "How Far Are We from the Planetary Health Diet? A Threshold Regression Analysis of Global Diets"

_foods, 2022, doi:10.3390/foods11070986_

Round 1

Reviewer 1 Report

Studying the influence of living conditions (in this case, GDP) on the behavioral characteristics of the population (in this case, food) is interesting. At the same time, in the article “How far are we from the Planetary Health Diet? A threshold regression analysis of global diets” has a number of points that require serious clarifications or corrections.

  1. The hypothesis of the article is the impact of the economic development of countries on the nutrition of the population. Obviously, on a global scale, eating is a habit, largely shaped by tradition and culture. If, for example, for the Dutch, the "traditional" dietary pattern is a high frequency of consumption of red meat and potatoes, along with a low frequency of consumption of low-fat dairy products and fruits (Van Dam RM et al. Patterns of food consumption and risk factors for cardiovascular disease in the general Dutch population. Am J Clin Nutr. 2003;77(5):1156–63.), then for, for example, in India, fruit-vegetable-cereal dietary patterns are "traditionally" preferred (Green R et al. Dietary patterns in India: a systematic review. Br J Nutr. 2016;116(1):142–8.). Therefore, perhaps it is not entirely correct to mix all the countries of the world in the analysis of the impact of economic development on nutrition? Or, alternatively, it is worth trying to stratify the analysis by groups of countries with different food cultures. It is likely that GDP has an impact on the consumption of food groups, however, not taking into account cultural traditions can lead to incorrect conclusions. I think that the authors should seriously change the analytical plan of the article in order to take this point into account.
  2. The introduction, in my opinion, is too long. For example, the text of lines 97-120 can be completely painlessly removed. At the same time, it is necessary to at least briefly describe the essence of the EAT-Lancet diet, which is often mentioned in the article, but not described.
  3. The methodology for calculating the total score is unclear (lines 169-175, <1 and >1), where it comes from and what it actually means.
  4. Agricultural land per capita, Level of population aging, Human Development Index are described as explanatory variables. Further in the text, these variables are not mentioned anywhere and it is not clear why and how they are considered.
  5. Lines 201-205 (Control variable) completely repeat lines 194-198 (Threshold variable).
  6. There are no references to figures in the text. The titles of the figures are more like notes to the figures.

Author Response

Thank you very much for your insightful comments! Please see the attachment.

Reviewer 2 Report

  1. General observation:
  • The topic is interesting, but the paper needs deep and major revisions.
  • Authors should apply Foods references style.
  • English editing is needed. Some of the sentences are complicated to understand. Please review the paper thoroughly and improve the quality of the writing.
  1. Introduction
  • The references and the numbers mentioned (undernourished, greenhouse gas emissions, etc.) in the introduction are outdated (you are using numbers from 2002, 2007, 2012, we are in 2022!). You need to update the numbers with new references.
  • After line 96, add details about the nutritious quality of the “western-style food consumption patterns”.
  • Line 60: Add (FAO) after The Food and Agriculture Organization of the United Nations.
  • Line 103, add Yams, millet, sweet potatoes.
  • Sentence 112-113, rice consumption in Asia is already high; please clarify.
  • Paragraph 115-120 is not clear.
  • Lines 107-114: add the references.
  • Explain why you use the Planetary Health Diet (PHD)? add more reference and information about it, its critics….
  • Add hypothesis.
  • The need for the research (i.e. identified gap) was not clearly explained or justified in the introduction.
  1. Results
  • Line 240, explain why you choose those 7 countries.
  • Add a table at the beginning of the results section to summarize your results by food group.
  1. Discussion
  • This section needs a major revision.
  • “The purpose of the discussion section is to interpret and describe the significance of your findings in relation to what was already known about the research problem being investigated”. So, here you should discuss the main evidence of the results and link it to the knowledge in the field.
  • Compare findings with literature review on topics presented in the introduction.
  1. Conclusion
  • Summarize or wrap up the main points in the paper.
  • Summarize your main results.
  • Add a paragraph to link your findings with the existing literature
  • Add a paragraph to propose future research avenues.
  • the conclusion has not highlighted the significance of the study or made any recommendations for how this could be advanced by further research or how it can be useful for policy, for instance
  1. References
  • Be sure that all references mentioned in the Reference List are cited in the text, and vice versa, including any references you may add.

Author Response

(The authors gave the same response as above.)

Round 2

Reviewer 1 Report

I am satisfied with the changes the authors have made. A few words about my first remark and the authors' response to it. I agree with the authors that isolating the contribution of cultural and traditional factors to the nutrition of the population is a difficult scientific task. The underestimation of these features significantly impoverishes the study and gives a somewhat one-sided idea of ​​the causal features of nutrition. Within the framework of the article under consideration, I, perhaps, agree with the sufficiency of indicating this as a limitation. However, if the authors continue their analysis in this direction, they may get interesting results. As a piece of advice, I can suggest introducing the “world regions” variable and its interaction with GDP into the regression models. It is quite possible that in different regions of the planet the associations of nutrition with GDP will differ. You can also try to consider the interaction with the political, ethnic, confessional characteristics of countries. It is possible to proceed from the opposite, that is, using data dimensionality reduction methods (cluster or factor analysis) to determine measures of similarity/differences between countries, followed by an assessment of associations with socio-demographic, cultural, and other characteristics of countries.

Reviewer 2 Report

Accept in the actual form.